# The Win–Win Effects of an Invasive Plant Biochar on a Soil–Crop System: Controlling a Bacterial Soilborne Disease and Stabilizing the Soil Microbial Community Network

**DOI:** 10.3390/microorganisms12030447

**Published:** 2024-02-22

**Authors:** Sheng Wang, Lei Wang, Sicong Li, Tiantian Zhang, Kunzheng Cai

**Affiliations:** 1College of Natural Resources and Environment, South China Agricultural University, Guangzhou 510642, China; wangsheng@stu.scau.edu.cn (S.W.); kishi218@163.com (L.W.); 2021lsc@stu.scau.edu.cn (S.L.); 2Key Laboratory of Tropical Agricultural Environment in South China, Ministry of Agriculture and Rural Affairs, South China Agricultural University, Guangzhou 510642, China; 3Guangdong Provincial Key Laboratory of Eco-Circular Agriculture, South China Agricultural University, Guangzhou 510642, China; 4College of Horticulture, South China Agricultural University, Guangzhou 510642, China; 17863608557@163.com

**Keywords:** bacterial wilt, *Solidago canadensis* L., biochar, microbial network complexity, soil microbial activity

## Abstract

Biochar is increasingly being recognized as an effective soil amendment to enhance plant health and improve soil quality, but the complex relationships among biochar, plant resistance, and the soil microbial community are not clear. In this study, biochar derived from an invasive plant (*Solidago canadensis* L.) was used to investigate its impacts on bacterial wilt control, soil quality, and microbial regulation. The results reveal that the invasive plant biochar application significantly reduced the abundance of *Ralstonia solanacearum* in the soil (16.8–32.9%) and wilt disease index (14.0–49.2%) and promoted tomato growth. The biochar treatment increased the soil organic carbon, nutrient availability, soil chitinase, and sucrase activities under pathogen inoculation. The biochar did not influence the soil bacterial community diversity, but significantly increased the relative abundance of beneficial organisms, such as *Bacillus* and *Sphingomonas*. Biochar application increased the number of nodes, edges, and the average degree of soil microbial symbiotic network, thereby enhancing the stability and complexity of the bacterial community. These findings suggest that the invasive plant biochar produces win–win effects on plant–soil systems by suppressing soilborne wilt disease, enhancing the stability of the soil microbial community network, and promoting resource utilization, indicating its good potential in sustainable soil management.

## 1. Introduction

*Ralstonia solanacearum* is a widely distributed soil-borne pathogen infecting over 200 host plants, which causes bacterial wilt disease and negatively impacts crop growth and agricultural sustainability [1]. Its hosts include tomato, potato, tobacco, banana, and various other economically important crops [2]. This pathogen invades plants by accumulating extracellular polysaccharides and secreting cell-wall-degrading enzymes in the plant xylem, causing wilting and substantially reducing crop yield [2,3]. Bacterial wilt is strongly influenced by climate, soil, and crop types. Effective prevention and control measures are still lacking that could be universally applicable to all crops on a sustainable basis [1,4]. Nowadays, different cultural, chemical, and biological approaches are being used for disease control but still have several limitations [3,5].

The potential of biochar as a soil amendment has rapidly increased, with diverse applications in the fields of environment, energy, agriculture, and industry [6,7]. Biochar positively influences plant growth, nutrient uptake, and soil microbial communities [8,9,10]. Studies reported that biochar promoted crop growth and enhanced resistance to various diseases, such as bacterial wilt in tomato and tobacco [11] and late blight in potato [12]. Different types and doses of biochar have been found effective in disease management, such as wood chip biochar (600 kg·hm^−2^) and bacterial wilt and black rot in tobacco [11,12,13,14,15], corn stover biochar (20 g·kg^−1^), and root rot in soybean [13]. The mechanisms underlying the biochar-mediated enhancement in plant resistance against soil-borne diseases are associated with the induction of plant resistance, the amelioration of the soil physicochemical properties, and the modulation of the soil microbial community structure [16,17].

Canadian goldenrod (*Solidago canadensis* L.), a native species from North America, has emerged as an invasive species in Southeastern China [18,19]. Its rapid reproductive abilities and widespread distribution make it a major ecological threat, impacting biodiversity, ecosystem functions, and the global economy [19,20]. Presently, the standard prevention practices of invasive species encompass mechanical removal, biological control, and chemical interventions. Although biological control targets specific species, it may be slow to produce effects and there exists risks of the invader developing resistance [18]. Chemical controls are effective, but result in detrimental environmental consequences [20]. Mechanical methods impose high economic costs and leave a significant amount of plant residues [21]. Therefore, transforming these residues into biochar presents a viable approach to managing invasive plants and controlling the diseases associated with them.

This study aims to find a sustainable method to utilize invasive plants without promoting their spread. By transforming these plant residues into biochar, this study aims to harness invasive plants for biochar production, evaluate its efficacy in managing a particular soil-borne disease, and investigate its influence on the structure of the soil microbial communities. Such an approach allows for an in-depth assessment of biochar’s impact on plant health and soil quality. Our hypotheses are: (1) biochar derived from invasive plants can efficiently mitigate soil-borne plant diseases; and (2) invasive plant biochar can improve the soil microbial community composition and increase the stability of the microbial network.

## 2. Materials and Methods

### 2.1. Plant Materials and Experimental Soil Conditions

The invasive plant *Solidago canadensis* L. was collected from Changsha, Hunan Province, China, at Mopanzhou. The pathogen *Ralstonia solanacearum* (strain Tim17) was obtained from the team of Professor Yuan Gaoqing at Guangxi University, Nanning, China. Tomato seeds/Jinfeng No.1 were purchased from Guangzhou Qiannong Agricultural Science and Technology Development Co., Ltd. (Guangzhou, China). The experimental soil was collected from a tomato field in Zhucun, Zengcheng City, Guangdong Province (113.70° E, 23.28° N). The basic physicochemical properties of the soil were as follows: organic matter of 16.2 g·kg^−1^, available nitrogen of 114 mg·kg^−1^, available phosphorus of 150 mg·kg^−1^, available potassium of 83 mg·kg^−1^, total nitrogen of 0.9 g·kg^−1^, total phosphorus of 1.2 g·kg^−1^, total potassium of 24.5 g·kg^−1^, and pH of 5.8.

### 2.2. Biochar Preparation

The collected Canadian goldenrod (*Solidago canadensis* L.). plant samples underwent several rinses with distilled water, followed by dehydration at 80 °C for 48 h to achieve desiccated biomass. Subsequently, the desiccated biomass was finely pulverized and passed through a 100-mesh sieve before undergoing pyrolysis in a muffle furnace under a nitrogen atmosphere, maintaining the temperature at 450 °C for 2 h intervals. The basic physicochemical properties of the invasive plant biochar were as follows: pH of 10.89, EC of 0.17 ms·cm^−1^, CEC of 26.63 cmol^+^·kg^−1^, C of 73.48%, H of 3.33%, O of 14.25%, N of 0.51%, and S of 0.22%.

### 2.3. Growing Conditions

Tomato seeds underwent surface sterilization in a 10% hydrogen peroxide (H_2_O_2_) solution for 10 min, followed by a series of three rinses in ultrapure water [22]. Post-sterilization, the seeds were incubated at 30 °C for 48 h in Petri dishes, each lined with a double layer of filter paper, to facilitate germination. The emergent seedlings were then relocated to plug trays filled with Klasmann-sourced sterilized peat soil (klasmann-Deilmann, Geeste, Germany). These trays were incubated in a controlled environment chamber (Model MGC-400B, Yiheng, Shanghai, China) set to maintain a constant temperature of 30 °C, relative humidity of 80%, and a photoperiod of 12 h light/12 h darkness. The seedlings received daily irrigation with a half-strength, tomato-specific nutrient solution, composed of 5 mM Ca (NO_3_)_2_, 1.88 mM K_2_SO_4_, 1.63 mM MgSO_4_, 0.5 mM KH_2_PO_4_, 0.04 mM H_3_BO_3_, 0.001 mM ZnSO_4_, 0.001 mM CuSO_4_, 0.01 mM MnSO_4_, 0.00025 mM Na_2_MoO_4_, 0.05 mM NaCl, and 0.1 mM Fe-EDTA. Upon attaining the third-leaf stage, the seedlings were transplanted to pots (16.5 cm × 17 cm), each containing 2 kg of the aforementioned sterilized peat soil, at a density of two seedlings per pot. To ensure the consistency and reproducibility of the experimental conditions, all pots were kept within the same controlled environment chamber throughout the study.

### 2.4. Experimental Design

The experiment was laid out in a completely randomized design with six treatments with three replicates of each, including 0%, 1%, and 2% biochar applications without (CK, SC1, and SC2) or with pathogen inoculation (CR, SR1, and SR2) The doses of biochar application were selected based on the results of our preliminary experiments and related references [11,12,13]. Each replicate had 40 tomato plants.

Once the tomato plants reached the sixth leaf stage, they were used to be inoculated with the Tim 17 strain of *R. solanacearum* using the root injury method. These strains were cultured on a LB medium, comprising 1 g·L^−1^ casamino acids, 10 g·L^−1^ difco peptone, and 1 g·L^−1^ yeast extract, and incubated at 30 °C for 48 h. Subsequently, the bacterial cells were harvested and underwent triple centrifugation in double-deionized water. The *R. solanacearum* suspension was carefully resuspended in deionized water, ensuring a homogeneous mixture. The concentration of this suspension was measured with a Shimadzu UV-2600 spectrophotometer (Kyoto, Japan), which operates at a wavelength of 600 nm. An optical density (OD) of 0.06 was recorded, equating to approximately 10^8^ colony-forming units per milliliter (cfu·mL^−1^). For initiating the infection, 50 mL of this bacterial suspension was applied uniformly to each pot. The disease progression in the tomato plants was rigorously documented, starting from the initial leaf wilt and continuing over a 15-day observation period. Moreover, plant and soil samples were collected, and the 1% biochar addition concentration with the lowest bacterial wilt disease index was selected for further analysis after 15 days of pathogen inoculation.

### 2.5. Disease Evaluation and Plant Growth

Destructive sampling, plant height, and dry weight were measured 15 days after inoculation with *R. solanacearum* when the disease index of non-biochar-added plants reached 80%. A disease index analysis was performed daily after inoculation according to the method of [23]. A rating of 0 was assigned when there were no observable symptoms, while ratings of 1 to 4 corresponded to the presence of one, two, three, or four wilted leaves, respectively. A rating of 5 signified the complete demise of the plant.

### 2.6. Soil Chemical Properties

The soil pH was determined using a water–soil suspension at a ratio of 2.5:1, employing an ST 2100 pH meter (Ohaus Instruments Co., Ltd., Changzhou, China) for the measurement. The electrical conductivity (EC) of the soil was quantified in a 5:1 water–soil solution utilizing an EC meter (ZDS-EC). The quantification of the soil organic carbon (SOC) was conducted via a Vario TOC cube elemental analyzer (Elementar Analysensysteme GmbH, Langenselbold, Germany). The determination of alkali-hydrolyzable nitrogen was performed using the alkali diffusion method. For the measurement of the available phosphorus (AP), the method involving 0.03 mol·L^−1^ NH_4_F and 0.025 mol·L^−1^ HCl was employed, while the available potassium (AK) was quantified using the NH_4_OAc-flame photometry method, as described by [24].

### 2.7. Soil Microbial Activity

The soil catalase activity was measured according to the potassium permanganate titration method [25]. The soil sucrase activity was determined using the 3,5-dinitrosalicylic acid colorimetric method [26]. The soil chitinase activity was measured using the colorimetric method according to the method of [27].

### 2.8. Determination of R. solanacearum Abundance in Soil

A quantitative real-time PCR, employing specific primers, was utilized to assess the soil abundance of *R. solanacearum*. The employed primer sequences were RSF (5′-TTCCTGGCTCAGATTGAACGC-3′) and RSR (5′-TGGTTACCTTGTTACGACTTCAC-3′). The PCR reaction mixture consisted of 10 μL 2× Green qPCR Master Mix, 0.5 μL of each primer (forward and reverse), 2 μL of DNA template, 2 μL of 10× Low ROX dye, and 5 μL of H_2_O. The thermal cycling conditions included an initial denaturation at 95 °C for 2 min, followed by 40 cycles of denaturation at 95 °C for 15 s and annealing at 60 °C for 15 s.

### 2.9. Analysis of the Soil Bacterial Community Structure

The soil microbial DNA was isolated utilizing the E.Z.N.A. Soil DNA Kit (Omega Bio-tek, Norcross, GA, USA), adhering stringently to the procedures specified by the manufacturer. The subsequent assessment of DNA integrity and purity was conducted via 1.0% agarose gel electrophoresis and quantification using a NanoDrop2000 spectrophotometer (Thermo Scientific, Waltham, MA, USA). The amplification of the V3-V4 regions of the 16S rRNA gene was achieved using designated primers (341F: 5′-CCTACGGGNGGCWGCAG-3′; 806R: 5′-GGACTACHVGGGTATCTAAT-3′). Following this, the purified amplicons were collectively sequenced on an Illumina platform, employing equimolar and paired-end sequencing (PE250), in accordance with the established protocols of Majorbio Bio-Pharm Technology Co., Ltd. (Shanghai, China). The raw sequencing reads were deposited into the NCBI Sequence Read Archive (SRA) database (Accession Number: PRJNA1073285).

### 2.10. Statistical Analysis

The data underwent analysis using the SPSS 26.0 software (SPSS, Chicago, IL, USA), with the application of a Duncan’s test to assess the significanc of the differences between the treatments post-harvest. All graphical figures were subjected to analysis and processing using GraphPad Prism 10.1.0 (GraphPad Software, San Diego, CA, USA). We visualized the structure of the bacterial communities based on the Bray–Curtis dissimilarity matrix (Vegan package). A redundancy analysis (RDA) was utilized to dynamically illustrate the relationships between environmental factors, samples, and communities. The Mantel test was employed to analyze the correlation between the environmental factors and the microbial communities. A Pearson’s correlation analysis was conducted to ascertain the linear correlations between the different parameters. The first principal components of the soil chemical properties, bacterial community diversity, and structure were used for further analysis.

## 3. Results

### 3.1. Plant Growth and Disease Control

Regardless of *R. solanacearum* inoculation, the 1% biochar application significantly increased plant height or dry weight, whereas the 2% biochar application negatively impacted plant growth (Figure 1). The biochar application presented a significant suppressive effect on wilt disease. The SR1 and SR2 treatments reduced the disease index of bacterial wilt by 56.0% and 28.3% at 10 days post-infection (dpi) and 49.2% and 14.0% at 15 dpi, respectively. Both the SR1 and SR2 treatments significantly reduced the abundance of *R. solanacearum* in the soil by 32.9% and 16.8% at 15 dpi, respectively. In addition, the 1% biochar application significantly increased the concentrations of plant P (45.1–46.2%) and K (15.8–16.1%) compared to that of CK, but had no significant effects on plant N concentrations (Appendix A).

### 3.2. Soil Chemical Properties

The biochar application improved the soil chemical properties, regardless of pathogen inoculation (Table 1). Under *R. solanacearum* inoculation, the 1% biochar (SR1) application increased the soil pH by 6.7%, organic carbon (SOC) by 54.4%, available phosphorus (AP) by 33.9%, available potassium (AK) by 69.5%, and EC by 96.2%. However, the biochar did not influence the soil available nitrogen (AN).

### 3.3. Soil Enzyme Activity

The biochar application significantly enhanced the soil enzyme activity, regardless of pathogen inoculation (Figure 2). The application of 1% biochar increased the activities of soil catalase by 7.2%, chitinase by 57.8%, and sucrase by 123.1% under non-*R. solanacearum* inoculation, and 21.1%, 94.4%, and 153.6%, respectively, under *R. solanacearum* inoculation. The correlation analysis showed that there was a significant negative correlation between the abundance of *R. solanacearum* in the soil and soil catalase (R^2^ = 0.969, *p* = 0.0004), chitinase (R^2^ = 0.991, *p* < 0.0001), and sucrase activities (R^2^ = 0.987, *p* < 0.0001) (Appendix A).

### 3.4. Soil Bacterial Community Structure and Diversity

The PCA analysis revealed that the biochar treatments led to distinct bacterial genus-level separations, regardless of *R. solanacearum* inoculation. The two principal components explained 47.91% and 51.38% of the total variance, respectively (Figure 3). Under non-inoculation conditions, as indicated by the PC1 coordinates, the CK and SC1 treatments had similar structures (Figure 3A). In contrast, under *R. solanacearum* inoculation, the SC1 treatment showed a unique bacterial community structure compared to the other treatments (Figure 3B). The analysis of all 12 samples (6 infected and 6 non-infected) highlighted notable differences between the bacterial communities under non-inoculation and *R. solanacearum* inoculation, underscoring the biochar’s significant influence on the soil bacterial communities (Appendix A).

Ten bacterial genera, including Bacills, Vicinamibacterales, Vicinamibacteraceae, Sphingomonas, Arthrobacter, noKD4-96, Gaiellales, Pseudolabrys, and Gemmatimonadaceae, were selected with > 0.1% relative abundance (Figure 4). The invasive plant biochar significantly influenced their relative abundances. Under *R. solanacearum* inoculation, the 1% biochar (SR1) treatment significantly increased the relative abundance of the soil beneficial microorganisms, including *Bacillus* (37.8%) and *Sphingomonas* (36.1%). Additionally, the four treatments shared 3,138 bacterial operational taxonomic units (OTUs) in the soil samples (Appendix A). The biochar treatments did not significantly affect the diversity of the soil bacterial community (Appendix A).

### 3.5. Co-Occurrence Network of Bacteria Analysis

The nodes in the symbiotic network represent various bacterial communities in the soil, whereas the edges signify their interactions. The application of 1% biochar exhibited a complex microbial community co-occurrence network compared to the non-biochar application (Figure 5A,B). The 1% biochar treatment significantly increased the averages, network edges, and nodes by 19.1%, 3.0%, and 2880.7%, respectively, under non-pathogen inoculation, and 15.4%, 2.6%, and 3.4%, respectively, under pathogen inoculation (Figure 5C–E). Furthermore, at the same biochar level, there was no significant effect on the average number of network edges and nodes, regardless of pathogen inoculation (Figure 5C–E).

### 3.6. Environmental Factors Influencing the Soil Mibiobial Community and Disease Index

The redundancy analysis (RDA) revealed species–environment relationships. The main variations were explained by the first and second axes, accounting for 67.55% and 15.70%, respectively. A positive correlation existed between the soil pH, catalase, chitinase, sucrase, and *Bacillus* and *Sphingomonas* abundances (Figure 6A). Furthermore, the abundance of *R. solanacearum* in the soil and the disease index of bacterial wilt showed a negative correlation with the soil pH, AK, SOC, and catalase, chitinase, and sucrase activities (Figure 6B).

## 4. Discussion

### 4.1. Invasive Plant Biochar Promotes Plant Growth and Inhibits Bacterial Wilt

As a potentially environmentally friendly agricultural material, biochar has been found to significantly inhibit soil-borne diseases in continuously cropped soils [28]. In our study, both the 1% and 2% *S. canadensis* L. biochar applications significantly suppressed bacterial wilt (Figure 1C). A related study found that the oil extracts of *S. canadensis* L. showed antibacterial properties on several phytopathogenic bacteria [29]. While the observed effect of biochar in suppressing bacterial wilt in our study was likely attributed more to biochar’s role in improving the soil environment than to the direct action of *S. canadensis* L., the disease-suppressing effects of biochar can vary due to differences in production conditions, raw materials, application ratios, and the types of diseases [14,15]. Previous studies have shown that coconut shell biochar at 10% and 30% significantly reduced the incidence of asparagus wilt disease [30], and oak biochar at 1% and 3% effectively mitigated botrytis cinerea stress in strawberries [31].

Furthermore, biochar can promote plant growth and nutrient absorption. Our results showed that 1% biochar increased the dry weight and height of tomato plants (Figure 1) and promoted phosphorus and potassium absorption (Appendix A), therefore maintaining plant health under pathogen inoculation. Studies showed that biochar application could increase the soil pH and ion exchange and improve soil structure, thereby facilitating plant nutrient absorption [32]. Increased nutrient levels may enhance plant resilience to ecological stresses and diseases. Roberts et al. [33] demonstrated that higher phosphorus levels promoted root system development and increased plant resistance against diseases, while the potassium supply reinforced cell walls and increased mechanical strength, thereby enabling a better resistance to environmental pressures and pathogen attacks [34]. In summary, biochar derived from invasive plants not only effectively suppressed bacterial wilt, but also promoted plant growth and nutrient absorption. Furthermore, it facilitated the reuse of resources from invasive plants.

### 4.2. Invasive Plant Biochar Stabilized the Soil Bacterial Co-Occurrence Network

The diversity and composition of the soil microbial community are critical to maintain the functional performance and ecological environment stability in the soil [35]. Our results reveal that the application of 1% biochar significantly increase the related abundance of bacterial genera, such as *Bacillus* and *Sphingomonas*, under pathogen inoculation (Figure 4). This finding is consistent with those of Wang et al. and Jin et al., who found that biochar increases the abundance of beneficial bacteria, thereby enhancing the competence with pathogens like *R. solanacearum* [36,37]. Furthermore, our study found that the abundance of *R. solanacearum* in the soil was significantly and negatively correlated with the abundance of *Bacillus* and *Sphingomonas* (Figure 6A). Prior studies indicated that biochar enhanced soil health through improving the microbial community structure [38,39]. However, our study found that the biochar application did not have any influence on the Shannon index, Simpson index, ACE index, and Chao index (Appendix A), which may be due to the biochar promoting the growth of specific bacteria while having a lesser impact on other species, therefore not altering the soil bacterial diversity [40,41].

The co-occurrence networks of microbial communities reflect the responses of soil microbial communities to environmental changes [42]. The stability of microbial co-occurrence networks has been considered a fundamental parameter to evaluate the ecosystem services and potential productivity [43,44]. Our results reveal that the proportion of Acidobacteriota in the soil co-occurrence network is the highest among all treatments, and biochar application leads to a more complex co-occurrence network (Figure 5A,B). Yang et al. also found that Acidobacteriota had an effect on plant growth and health, such as promoting plant growth, enhancing nutrient absorption, and increasing plant resistance to adverse conditions [45]. In the present study, the biochar treatment significantly increased the number of nodes, average degree, and edges in the soil bacterial co-occurrence network (Figure 5). This finding indicates that biochar can potentially enhance the stability and complexity of the microbial community networks. Previous studies suggested that complex interactions among microbes were beneficial for the stability of the microbial community, thereby enhancing the resistance of the microbial community to pathogen invasion [46].

### 4.3. Invasive Plant Biochar Enhanced the Soil Quality and Mitigated Diseases in Tomato Plants

The soil quality (the physical, chemical, and microbial properties of soil) is a significant factor affecting the incidence of plant diseases in the soil subjected to continuous cropping, thereby impacting the abundance of bacterial pathogens in the soil [4,47]. The soil pH is considered one of the vital indicators of soil health, with a low pH stimulating the activity of *R. solanacearum* in the soil and influencing plant disease incidence [48,49]. In this study, the application of invasive plant-derived biochar resulted in a substantially elevated soil pH (Table 1), which was similar to the results of other studies [28,50]. Furthermore, a substantial negative correlation was observed between the soil pH and the abundance of *R. solanacearum* (Figure 6B). This result is consistent with that of a previous study by Li et al., who found that soil acidification exacerbated bacterial wilt incidence [48]. Notably, pH significantly affected the composition of the microbial communities, displaying a positive correlation with the relative abundance of *Bacillus* (Figure 6A). Furthermore, biochar’s improvements in soil quality, such as increased organic carbon and nutrient availability (Table 1), may have also contributed to these observed changes in the microbial community composition (Figure 6), thereby reducing soil-borne diseases in tomato plants, as shown in previous studies [48,50]. 

Our results also reveal that the biochar application significantly increased soil sucrase, catalase, and chitinase activities. Research indicated that the enhanced activity of soil enzymes led to an improved soil structure and accelerated nutrient cycling, which further facilitated plant growth [51]. It was reported that the enhancement in the soil microbial activity could inhibit the occurrence of soil-borne diseases [52]. A correlation analysis revealed substantial inverse correlations of the abundance of *R. solanacearum* with the activities of chitinase, catalase, and sucrase in the soil (Figure 6 and Appendix A). These correlations indicate that the soil enzyme activity plays an important role in suppressing pathogens. Chitinase enzyme degrades chitin and plays a crucial role in plant defense by impeding the growth of pathogens via cell wall disintegration, while also boosting the plant’s immune system and improving its resistance to pathogens [53,54]. Sucrase and catalase modify the nutrient levels and redox conditions in the soil, creating an unfavorable environment for the survival of pathogens [55,56]. Additionally, these enzymes not only enhance soil fertility by promoting organic matter decomposition and nutrient cycling in the soil, but also indirectly support healthier plant growth and increase the resistance of plants to pathogens [53,54].

## 5. Conclusions

The application of invasive plant biochar significantly reduced the abundance of *R. solanacearum* in the soil and effectively decreased the incidence of bacterial wilt, providing an effective approach to mitigate a particular soil-borne bacterial disease. The application 1% biochar significantly influenced the composition of the soil microbial community through increasing the abundance of beneficial bacteria, such as *Bacillus* and *Streptomyces*. Additionally, the 1% biochar increased the soil pH and promoted tomato growth, while the pH increase could be a significant factor in reducing the abundance of *R. solanacearum* in the soil. The application of invasive plant biochar also led to the increase in the number of nodes, edges, and the average degree of microbial symbiotic networks, consequently enhancing the stability and complexity of the bacterial community. These cumulative effects have a positive impact on the stability and sustainability of the studied soil system. Our findings suggest that the application of an invasive plant biochar results in mutually beneficial outcomes within the studied soil–crop system under controlled conditions.

## Figures and Tables

**Figure 1 microorganisms-12-00447-f001:**
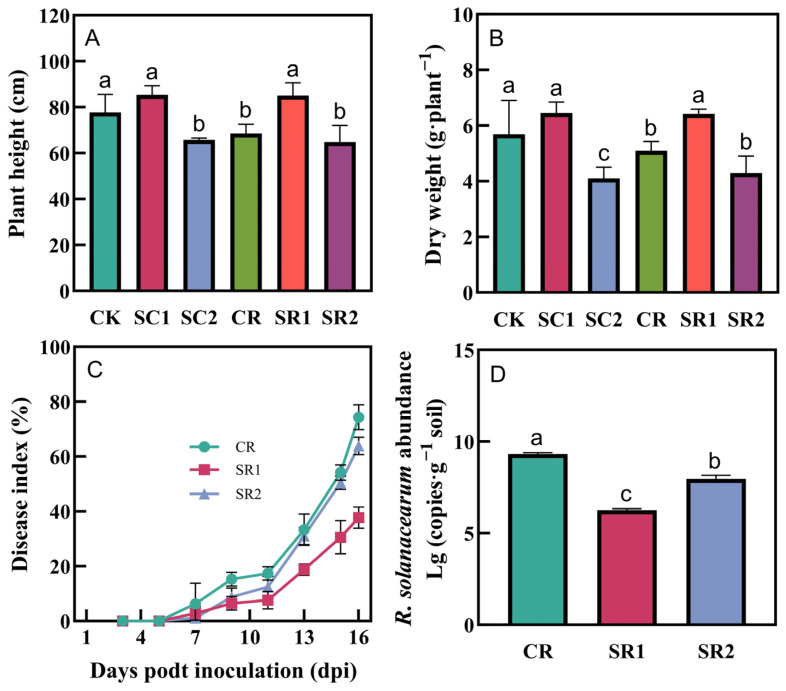
Effects of the invasive plant biochar treatments on plant height (**A**), dry weight(**B**), (**C**) disease index, and (**D**) *R. solanacearum* abundance in soil. CK, SC1, and SC2 represent 0%, 1%, and 2% biochar applications without pathogen inoculation, respectively; CR, SR1, and SR2 represent 0%, 1%, and 2% biochar applications with pathogen inoculation, respectively. All values are presented as the mean ± standard error (*n* = 3). Different letters in the treatments indicate significant differences (*p* < 0.05).

**Figure 2 microorganisms-12-00447-f002:**
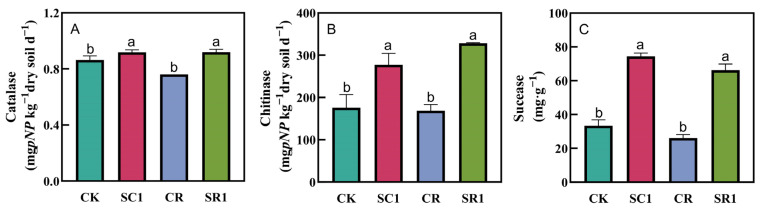
Effects of the biochar treatments on the soil enzymatic activities. (**A**) Catalase, (**B**) chitinase, and (**C**) sucrase. CK and SC1 represent 0% and 1% biochar applications without pathogen inoculation, respectively. CR and SR1 represent 0% and 1% biochar applications with pathogen inoculation, respectively. All values are presented as the mean ± standard error (*n* = 3). Different letters in the treatments indicate significant differences (*p* < 0.05).

**Figure 3 microorganisms-12-00447-f003:**
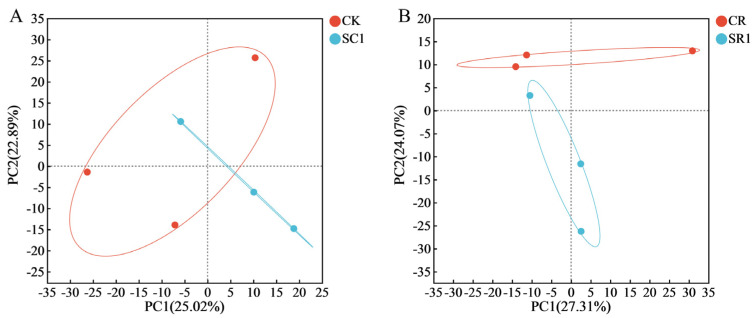
Effects of the invasive plant biochar treatments on the soil bacterial community structure. (**A**) non-infected, (**B**) infected. CK and SC1 represent 0% and 1% biochar applications without pathogen inoculation, respectively. CR and SR1 represent 0% and 1% biochar applications with pathogen inoculation, respectively.

**Figure 4 microorganisms-12-00447-f004:**
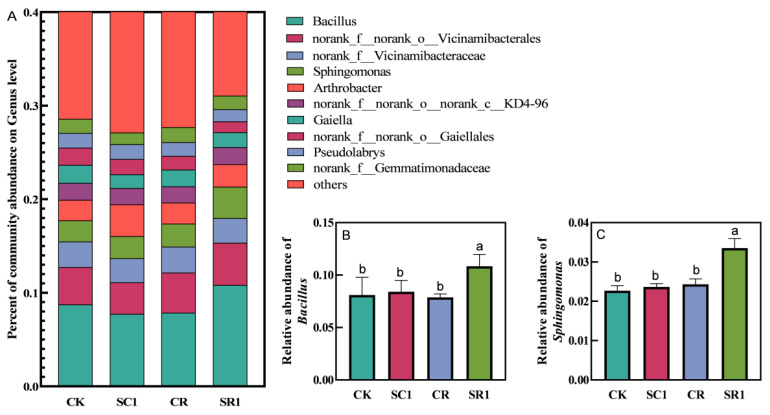
Effect of the invasive plant biochar treatments on the bacterial community composition in the soil. (**A**) Genus level, (**B**) *Bacillus*, and (**C**) *Sphingomonas*. CK and SC1 represent 0% and 1% biochar applications without pathogen inoculation, respectively. CR and SR1 represent 0% 1and % biochar applications with pathogen inoculation, respectively. All values are presented as the mean ± standard error (*n* = 3). Different letters in the treatments indicate significant differences (*p* < 0.05).

**Figure 5 microorganisms-12-00447-f005:**
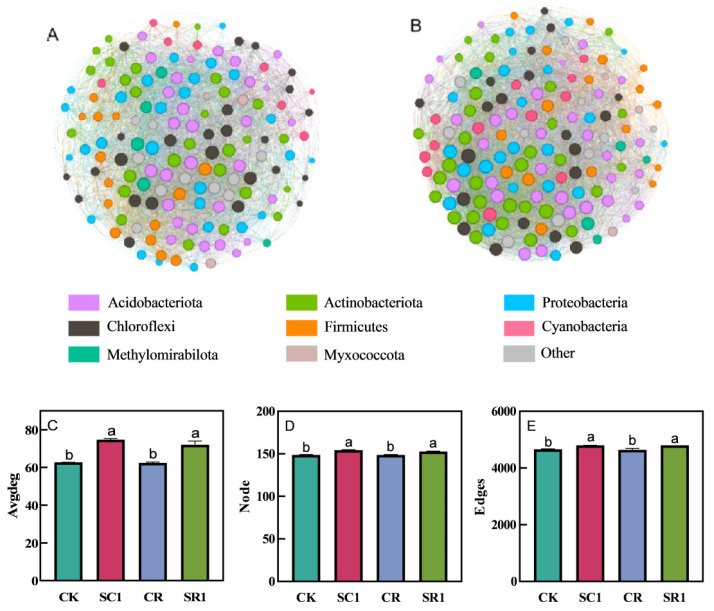
Effect of the invasive plant biochar treatments on the co-occurrence network analysis of the soil bacterial communities. Co-occurrence network in 0% biochar (CK and CR) (**A**), 1% biochar (SC1 and SR1) (**B**), average degree (**C**), node (**D**), and edge (**E**). CK and SC1 represent 0% and 1% biochar applications without pathogen inoculation, respectively. CR and SR1 represent 0% and 1% biochar applications with pathogen inoculation, respectively. All values are presented as the mean ± standard error (*n* = 3). Different letters in the treatments indicate significant differences (*p* < 0.05).

**Figure 6 microorganisms-12-00447-f006:**
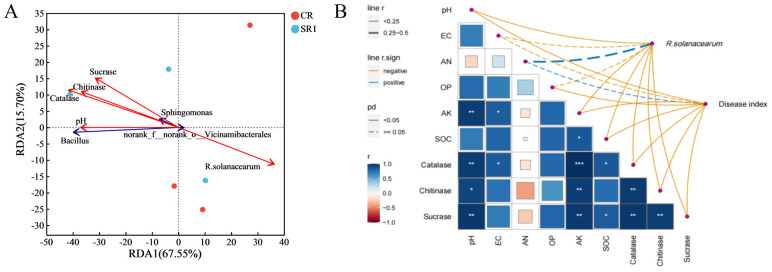
RDA analysis (**A**) and Mantel test (**B**) analysis of the environmental factors and bacterial community structure. CR and SR1 represent 0% and 1% biochar applications with pathogen inoculation, respectively. * *p* < 0.05, ** *p* < 0.01, *** *p* < 0.001.

**Table 1 microorganisms-12-00447-t001:** Effect of the invasive plant biochar treatments on the soil chemical properties.

Treatments	pH	EC (ds·cm^−1^)	SOC (g·kg^−1^)	AK (mg·kg^−1^)	AP (mg·kg^−1^)	AK (mg·kg^−1^)
CK	7.6 ± 0.1 b	77.3 ± 12.0 b	11.1 ± 1.4 b	47.8 ± 2.0 a	28.0 ± 4.1 b	135.2 ± 9.9 b
SC1	8.1 ± 0.0 a	113.7 ± 7.8 ab	21.5 ± 1.3 a	46.7 ± 2.0 a	39.3 ± 3.5 a	204.7 ± 10.7 a
CR	7.5 ± 0.1 b	83.3 ± 3.1 b	13.2 ± 3.6 b	52.5 ± 7.0 a	28.9 ± 6.6 b	129.9 ± 1.2 b
SR1	8.0 ± 0.1 a	161.0 ± 55.5 a	20.4 ± 1.8 a	50.2 ± 5.3 a	38.7 ± 1.1 a	220.2 ± 12.3 a

CK and SC1 represent 0% and 1% biochar applications without pathogen inoculation, respectively. CR and SR1 represent 0% and 1% biochar application with pathogen inoculation, respectively. All values are presented as the mean ± standard error (*n* = 3). Different letters in the treatments indicate significant differences (*p* < 0.05).

## Data Availability

The data and results of this study are available upon reasonable request. Please contact the main author of this publication.

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
