# Peer review of "The Win–Win Effects of an Invasive Plant Biochar on a Soil–Crop System: Controlling a Bacterial Soilborne Disease and Stabilizing the Soil Microbial Community Network"

_microorganisms, 2024, doi:10.3390/microorganisms12030447_

Round 1

Reviewer 1 Report

Comments and Suggestions for Authors

Comments:

The paperThe win-win effects of invasive plant biochar on soil-crop systems: Controlling soilborne disease and stabilizing soil microbial community network” by Wang et al. is interesting, comprehensive and focuses on an important thematic, which is the use of biochar and its potential benefits to plants.

The paper is well written but needs revision in terms of typos. I pointed out some examples, but other could have escaped to me. The paper is well organised and the study was well designed. I did like the study!

The tittle seems too forceful to me. The authors studied one plan-soil system only, not various, and tested one biochar (on a particular bacterial disease only), prepared from one invasive plant. Moreover, that particular system was studied in lab conditions and then in pots (by the way, the conditions where pots were placed are not described…). So the tittle should be, in my opinion, The win-win effects of an invasive plant biochar on a soil-crop system: Controlling a bacterial soilborne disease and stabilizing soil microbial community network”.

I do think that we cannot extrapolate to other systems. Please be cautious about this also in abstract, and elsewhere in text.

The Abstract is clear, concise, pointing out the main results. Please add “the” before “invasive plant” in line 16. Also, add a dot after “L.” in line 17.

Keywords: I would change some of them, as some are listed in the title, so do not repeating them in these both sections could be wise (title and keywords).

The Introduction is well written, and the objectives and hypotheses are clearly stated; the introduction is a good state of the art and points out why this study is needed and important. Nevertheless, in my opinion, the authors should state somewhere in Introduction that this study does not intend to increase or inspire the spreading of invasive plants. This is just a personal opinion.

But as I said, in line 65, I would change the objective to “in managing a particular soil-borne disease” (singular).

Materials and Methods are appropriate but they should be more well explained and described, at least in some subsections. Please see below. The used techniques and methodologies were adequate for the study, and the experimental design was appropriate. The statistical analyses were also adequate.

Section 2.1 please, refer the cultivar/genotype of tomato seeds/plants used. Also replace in the title of this subsection “experiment soil” by “experimental soil” (line 72) and also “Planting materials” by “Plant materials”.

Just a point: in each section/subsection/paragraph (even in subsections tittles) I prefer not to use abbreviations of scientific names in the first time these are cited. Only after the second use in the same subsection, you should abbreviate these. So, for example, in section 2.1, line 73, use Solidago canadensis instead of S. canadensis L.. Here and thereafter the name of the author is not needed anymore as it is already stated in Introduction. The same for the scientific name of the bacterium. Please that sometimes you wrote “L” from Linnaeus without a dot and it must be “L.”. But, as I told you, you do not need to repeat the author of the name after the first time. See for example line 84. Also please make a break between the name of the genus and the name (see lines 84 and 111). They are joined in these situations (here, as I mentioned before, I prefer the full names of the genus).

Section 2.3. The growing conditions of the plants in pots are not described. Were these kept on the same chamber? I believe so, as nothing different is said. But this has to be clearer to me and to the readers (lines 103-105). I guess that the plots were not transferred to a greenhouse.

Also, if possible, make it clearer the dimensions of the pots. I guess that one dimension is the height and the other is the diameter or width… If the pots are cubic or so, there is a dimension lacking. This might be an over-zealousness thing, but I think the information can be more accurate.

Section 2.4 Please explain why using that particular concentration of bacteria for infection. Is this the common procedure? Also, please describe better how the suspension of bacterial cells were applied. Were the 50 mL of this bacterial suspension applied uniformly to each pot, before or after the plants were transferred to pots? Was the suspension applied in the middle of the two plants or all over the soil of the pot, or at the basis of each plant (there were two tomato plants at the sixth-leaf stage per pot)? Please, describe what you mean by “uniformly”. This is not clear to me.

Also, in line 111, replace “to inoculated plants” by “to inoculate plants”. Moreover, did the authors use only one strain or several strains? As if you used several strains, you must be sure that they have the same virulence against tomato plants. I believe that the authors did use only one strain (strain Tim 17). Please make it clearer. One strain, not several strains (line 111). It should be “Ralstonia solanacearum strain (Tim 17) was used to inoculate…”

Section 2.5 Please add “Cheng and collaborators” before [22] (line 128). Moreover, I do think that the determination of the Disease index should be briefly explained in this section and not remitting only to [22] reference. This is also valid to sections 2.6 and 2.7. Brief explanations (at least brief descriptions) are important instead of only referring to particular published studies…

Moreover, why did not the authors determine the bacterial loads inside the plants (as they did for soil), besides performing the determination of disease index?

Section 2.9 Data of metagenomic analysis data should be deposited in a public database and this is not addressed in the paper.

Results are in general well and extensively described with the appropriate number of tables and figures and supplementary materials, but some of the figures/images need to be enlarged to become more comprehensive. There are some typos in Table 1 (number of lines are in table, somewhat merged in the first column).

Minor things: “Figure 1” is in italics in line 186. Lines 236, 237, 239 and 241: the names of the bacteria shall be in italics. Please correct.

Also, in line 238, replace “invaded” by “invasive”. The plant is an invasive one. The plant is not invaded. Also correct it (invasive and not invaded) in Conclusions (line 378). Also, see below in these “Comments”.

The authors use the term “nodes” (line 252 and also in abstract) and this can be unwise because of the term node of, for example, symbiotic or pathogenic infections in plants. Is there a way to use another term here, in this case, instead of nodes? For example, “intersection points”, that is exactly what is meant? It is a suggestion.

Section 3.6. The legend of figure 6 (figures 6A and 6B) is not missing but it is only shown in lines 276-278. It is wrongly placed. I would put the figure and its legend after the text of section 3.6, at the end of this section.

Discussion of the results is extensive and well organised in three main subsections. This is valuable to the discussion and makes it stronger. All of these subsections are supported by literature and results are compared.

The conclusions summarize the main results and achievements of the study, but are too much inflated. So, in line 369, I would change “an effective approach to mitigate soilborne diseases” to “an effective approach to mitigate a particular soilborne bacterial disease”. Also, in lines 376-378, I recommend changing “These cumulative effects have a positive impact on the stability and sustainability of soil ecosystems. This suggests that the selection and application of invaded plant biochar resulted in a mutually beneficial outcomes within the soil-crop system” to “These cumulative effects have a positive impact on the stability and sustainability of the studied soil system. This suggests that the selection and application of an invasive plant biochar resulted in mutually beneficial outcomes within the soil-crop studied system under laboratorial controlled conditions”. The written phrases in these lines had some errors, as you can realize…

The list of references is suitable (in terms of original research papers and reviews), and it supports well the paper, especially in its introduction and discussion, concerning the biochar utilization.

Comments on the Quality of English Language

English needs some revision, especially taking into account some typos that I could find (please see my comments). 

Author Response

Dear Prof.   :

Thank you very much for giving us an opportunity to revise our manuscript. We appreciate

the editor and reviewers very much for their constructive comments and suggestions on our

manuscript entitled “The win-win effects of an invaded plant biochar on a soil-crop system: Controlling a bacterial soilborne disease and stabilizing soil microbial community network”.

We have studied reviewers’ comments carefully. According to the reviewers’ detailed

suggestions, we have made a careful revision on the original manuscript. All revised portions are

marked in red in the revised manuscript which we would like to submit for your kind

consideration.

  1. So the tittle should be, in my opinion, “The win-win effects of an invasive plant biochar on a soil-crop system: Controlling a bacterial soilborne disease and stabilizing soil microbial community network”.

Response: Thank you for your suggestion. It has been replaced as suggested. in line 2-4.

  1. Please add “the” before “invasive plant” in line 16. Also, add a dot after “L.” in line 17.

Response: Thank you for your suggestion. It has been replaced as suggested.in line 16-17.

  1. Keywords: I would change some of them, as some are listed in the title, so do not repeating them in these both sections could be wise (title and keywords).

Response: Thank you for your suggestion. We have reconsidered the keywords and replaced "microbial community" with "Microbial network complexity" to better reflect the focus of our study on the intricate interactions within soil microbial networks following biochar application. This adjustment aims to highlight the significant findings related to the complexity and stability of microbial networks in soil ecosystems influenced by invasive plant biochar.in line 29.

  1. But as I said, in line 65, I would change the objective to “in managing a particular soil-borne disease” (singular).

Response: Thank you for your suggestion. It has been replaced as susted. in line 72.

  1. Section 2.1 please, refer the cultivar/genotype of tomato seeds/plants used. Also replace in the title of this subsection “experiment soil” by “experimental soil” (line 72) and also “Planting materials” by “Plant materials”.

Response: Thank you for your suggestion. It has been replaced as suggested. in line 79.

  1. Just a point: in each section/subsection/paragraph (even in subsections tittles) I prefer not to use abbreviations of scientific names in the first time these are cited. Only after the second use in the same subsection, you should abbreviate these. So, for example, in section 2.1, line 73, use Solidago canadensis instead of S. canadensis L.. Here and thereafter the name of the author is not needed anymore as it is already stated in Introduction. The same for the scientific name of the bacterium. Please that sometimes you wrote “L” from Linnaeus without a dot and it must be “L.”. But, as I told you, you do not need to repeat the author of the name after the first time. See for example line 84. Also please make a break between the name of the genus and the name (see lines 84 and 111). They are joined in these situations (here, as I mentioned before, I prefer the full names of the genus).

Response: Thank you for your suggestion. It has been replaced as suggested. in line 80-89.

  1. Section 2.3. The growing conditions of the plants in pots are not described. Were these kept on the same chamber? I believe so, as nothing different is said. But this has to be clearer to me and to the readers (lines 103-105). I guess that the plots were not transferred to a greenhouse.

Response: Thank you for your suggestion. It has been replaced as suggested. in line 112-114.

  1. Section 2.4 Please explain why using that particular concentration of bacteria for infection. Is this the common procedure? Also, please describe better how the suspension of bacterial cells were applied. Were the 50 mL of this bacterial suspension applied uniformly to each pot, before or after the plants were transferred to pots? Was the suspension applied in the middle of the two plants or all over the soil of the pot, or at the basis of each plant (there were two tomato plants at the sixth-leaf stage per pot)? Please, describe what you mean by “uniformly”. This is not clear to me.

Response: Thank you for your valuable comments and questions regarding our manuscript. We greatly appreciate your attention to the selection of the bacterial suspension concentration and its application method. Here, we provide detailed explanations and necessary revisions to address your concerns.

Selecting a specific concentration of Ralstonia solanacearum (108 CFU/mL) for plant infection experiments is based on previous literature and preliminary experimental results. This concentration is widely considered to be able to simulate the disease pressure under natural conditions, while ensuring the consistency and repeatability of disease occurrence, and is a common method used to study the impact of soilborne diseases on plants.

In response to your inquiries about the method of bacterial suspension application, we have revised the original text to clarify the uniform application approach. The bacterial suspension was applied uniformly to each pot after the tomato plants were transplanted, ensuring that the plants were in their final growth environment. We distributed 50 mL of the bacterial suspension evenly across the entire soil surface of the pot, rather than just in the middle of the two plants or at the base of each plant individually. This was done to ensure that every part of the soil, including the root zone of both plants, was equally exposed to the same concentration of the pathogen. By "uniformly," we mean the even distribution of the bacterial suspension over the soil surface, aiming to replicate the even presence of pathogens in a natural soil environment.

  1. Also, in line 111, replace “to inoculated plants” by “to inoculate plants”. Moreover, did the authors use only one strain or several strains? As if you used several strains, you must be sure that they have the same virulence against tomato plants. I believe that the authors did use only one strain (strain Tim 17). Please make it clearer. One strain, not several strains (line 111). It should be “Ralstonia solanacearumstrain (Tim 17) was used to inoculate…”

Response: Thank you for your suggestion. It has been replaced as suggested.in line 121-122.

  1. why did not the authors determine the bacterial loads inside the plants (as they did for soil), besides performing the determination of disease index?

Response:  Thank you for your suggestion. Our study focuses on the impacts of biochar on soilborne diseases and soil microbial communities, rather than on the internal colonization of plants by pathogens. The determination of disease index provides a direct measure of the disease's impact on plant health, which might have been deemed sufficient for the study's objectives.

  1. Section 2.9 Data of metagenomic analysis data should be deposited in a public database and this is not addressed in the paper.

Response: Thank you for your suggestion. It has been replaced as suggested.in line 180-181.

  1. Results are in general well and extensively described with the appropriate number of tables and figures and supplementary materials, but some of the figures/images need to be enlarged to become more comprehensive. There are some typos in Table 1 (number of lines are in table, somewhat merged in the first column).

Response: Thank you for your suggestion. It has been replaced as suggested.in line 218.

  1. Minor things: “Figure 1” is in italics in line 186. Lines 236, 237, 239 and 241: the names of the bacteria shall be in italics. Please correct.

Response: Thank you for your suggestion. It has been replaced as suggested. in line 257-262

  1. Also, in line 238, replace “invaded” by “invasive”. The plant is an invasive one. The plant is not invaded. Also correct it (invasive and not invaded) in Conclusions (line 378). Also, see below in these “Comments”.

Response: Thank you for your suggestion. It has been replaced as suggested. in line 266

  1. The authors use the term “nodes” (line 252 and also in abstract) and this can be unwise because of the term node of, for example, symbiotic or pathogenic infections in plants. Is there a way to use another term here, in this case, instead of nodes? For example, “intersection points”, that is exactly what is meant? It is a suggestion.

Response:Thank you for your meticulous review and valuable suggestions regarding the use of the term 'nodes' in our article. We understand your concern that this term may cause confusion, especially in the field of plant pathology where 'nodes' might be associated with symbiotic or pathogenic infections. However, 'nodes' is a widely accepted and used term in network analysis, specifically referring to individual entities or connection points within a network, whether in ecological networks, social networks, or other forms of complex system analysis.

In our study, the term 'nodes' is used to describe the connection points among various microbial species within the soil microbial community network. These 'nodes' represent microbial species, and the connections or edges between nodes represent their potential interactions, such as symbiosis, competition, or predation. This expression not only aligns with the standard terminology of current microbial ecology and network theory but also accurately reflects the core content and findings of our research.

Despite this, we deeply understand that the choice of terminology needs to consider clarity and unambiguity across different disciplinary backgrounds. Regarding the proposed alternative term 'intersection points,' while it can describe the relational position between individuals within a network to some extent, it does not possess the same level of professionalism and precision as 'nodes' in scientific and technical literature. Therefore, considering the consistency and widespread acceptance of professional terminology, we prefer to continue using the term 'nodes'.

  1. Section 3.6. The legend of figure 6 (figures 6A and 6B) is not missing but it is only shown in lines 276-278. It is wrongly placed. I would put the figure and its legend after the text of section 3.6, at the end of this section.

Response: Thank you for your suggestion. It has been replaced as suggested. in line 293-296

  1. In line 369, I would change “an effective approach to mitigate soilborne diseases” to “an effective approach to mitigate a particular soilborne bacterial disease”. Also, in lines 376-378, I recommend changing “These cumulative effects have a positive impact on the stability and sustainability of soil ecosystems. This suggests that the selection and application of invaded plant biochar resulted in a mutually beneficial outcomes within the soil-crop system” to “These cumulative effects have a positive impact on the stability and sustainability of the studied soil system. This suggests that the selection and application of an invasive plant biochar resulted in mutually beneficial outcomes within the soil-crop studied system under laboratorial controlled conditions”. The written phrases in these lines had some errors, as you can realize…

Response: Thank you for your suggestion. It has been replaced as suggested.in line 388-390

Reviewer 2 Report

Comments and Suggestions for Authors

Review Report: microorganisms-2865990

The study investigates the impact of biochar derived from the invasive plant Solidago canadensis L. on bacterial wilt control, soil quality, and microbial regulation. Overall, the findings are significant and contribute valuable insights into the complex interactions among biochar, plant resistance, and soil microbial communities.

The application of invasive plant biochar demonstrated a significant reduction in the abundance of Ralstonia solanacearum, contributing to a substantial decrease in wilt disease index (14.0% - 49.2%). This highlights the potential of invasive plant biochar in effectively controlling soilborne diseases. Furthermore, the study revealed a positive impact on tomato growth, indicating that the invasive plant biochar has a beneficial effect on plant health. This finding is crucial for understanding the holistic benefits of biochar application in sustainable agriculture. Biochar treatment led to increased soil organic carbon, improved nutrient availability, and elevated enzyme activities (chitinase and sucrase) under pathogen inoculation. These observations suggest that invasive plant biochar positively influences soil quality, fostering a conducive environment for plant growth. While biochar did not affect soil bacterial community diversity, it significantly increased the abundance of beneficial bacteria such as Bacillus and Sphingomonas. The study's focus on the microbial community provides insights into the mechanisms underlying biochar-induced changes in the soil ecosystem. The investigation of the soil microbial symbiotic network revealed that biochar increased the number of nodes, edges, and average degree, indicating improved stability and complexity of the bacterial community. This analysis adds depth to our understanding of how biochar influences soil microbial interactions.

Comments to authors

Clarification on Mechanisms: It would be beneficial if the authors could provide more insights into the specific mechanisms through which invasive plant biochar exerts its influence on disease suppression and microbial community dynamics. This could enhance the study's contribution to the existing literature.

Generalization: While the study focuses on the specific invasive plant Solidago canadensis L, discussing the potential applicability of the findings to other biochar sources and plant species could broaden the study's relevance.

The "Materials and Methods" section provides a detailed description of the experimental setup, including the collection of planting materials, biochar preparation, growing conditions, experimental design, disease evaluation, and various analytical techniques. While the section is generally comprehensive, there are some shortcomings that need to be addressed:

Plant Sterilization Method: The section mentions the surface sterilization of tomato seeds using a 10% hydrogen peroxide solution but lacks details on the rationale behind this choice. Providing a brief explanation or citing relevant literature would enhance the clarity of the methodology.

Biochar Application Concentrations: While the biochar application concentrations (0%, 1%, and 2%) are mentioned, the rationale behind selecting these specific concentrations is not provided. Including a brief explanation or referencing previous studies that justify these concentrations would add context to the experimental design.

Temporal Aspects of Disease Evaluation: The disease progression is monitored over a 15-day observation period, but the choice of this specific duration is not justified. Providing a rationale for the chosen timeframe or referring to literature supporting this decision would improve the methodological clarity.

Addressing these points will enhance the overall clarity, transparency, and rigor of the "Materials and Methods" section in the manuscript.

I recommend accepting this article for publication, pending minor revisions and additional clarification on the underlying mechanisms.

Overall Recommendation: Accept with Minor Revisions.

Author Response

Dear Prof. :

Thank you very much for giving us an opportunity to revise our manuscript. We appreciate the editor and reviewers very much for their constructive comments and suggestions on our manuscript entitled “The win-win effects of an invaded plant biochar on a soil-crop system: Controlling a bacterial soilborne disease and stabilizing soil microbial community network”.

We have studied reviewers’ comments carefully. According to the reviewers’ detailed

suggestions, we have made a careful revision on the original manuscript. All revised portions are marked in red in the revised manuscript which we would like to submit for your kind consideration.

  1. Plant Sterilization Method: The section mentions the surface sterilization of tomato seeds using a 10% hydrogen peroxide solution but lacks details on the rationale behind this choice. Providing a brief explanation or citing relevant literature would enhance the clarity of the methodology.

 Response: Thank you for your suggestion. We have added the following reference [22] as suggested.

  1. Liu, K.; Shen, L.; Sheng, J. Improvement in cadmium tolerance of tomato seedlings with an antisense DNA for 1-aminocyclopropane-1-carboxylate synthase. J Plant Nutr, 2008, 31(5), 809-827.[CrossRef]

  1. Biochar Application Concentrations: While the biochar application concentrations (0%, 1%, and 2%) are mentioned, the rationale behind selecting these specific concentrations is not provided. Including a brief explanation or referencing previous studies that justify these concentrations would add context to the experimental design.

 Response:Thank you for your valuable feedback. We acknowledge the need for clarity on our choice of biochar application rates (0%, 1%, 2%) and appreciate your suggestion. The selection was based on previous studies indicating these concentrations' effectiveness in soil quality improvement, plant growth promotion, and disease suppression. In our revision, we will succinctly justify these choices with appropriate references, providing a solid foundation for our experimental design. Your guidance is greatly appreciated as we refine our manuscript.

  1. Li, C.; Ahmed, W.; Li, D.; Yu, L.; Xu, L.; Xu, T.; Zhao, Z. Biochar suppresses

bacterial wilt disease of flue-cured tobacco by improving soil health and functional diversity of rhizosphere microorganisms. Appl. Soil Ecol. 2022, 171, 104314.[CrossRef]

  1. Jin, L.; Feng, S.; Tang, S.; Dong, P.; Li, Z. Biological control of potato late blight with a combination of Streptomycesstrains and biochar. Biol. Control. 2023, 183, 105248.[CrossRef]
  2. Rogovska, N.; Laird, D.; Leandro, L.; Aller, D. Biochar effect on severity of soybean root disease caused by Fusarium virguliforme. Plant Soil. 2017, 413(1-2), 111-126.[CrossRef]

3.Temporal Aspects of Disease Evaluation: The disease progression is monitored over a 15-day observation period, but the choice of this specific duration is not justified. Providing a rationale for the chosen timeframe or referring to literature supporting this decision would improve the methodological clarity.

Response: Thank you for your insightful comments, we have revised it. Our decision to monitor disease progression over a 15-day period was empirically based on preliminary observations indicating that, under inoculation conditions, the disease index in control treatments reached 80% by this time, as illustrated in Figure 3C.

Round 2

Reviewer 1 Report

Comments and Suggestions for Authors

Comments:

First of all, I must say that in spite of the authors state that my suggestions were taken into account, some errors still occur. For example, the authors keep using the expression “invaded plant” and they should use “invasive plant” (or “invading plant” or “plant invader” or “invasive alien plant” or “alien invader”). We are talking about an invasive plant (or plant invader), not a plant that is invaded! This error occurs all through the whole manuscript (36 times), namely in the tittle, keywords, introduction, materials and methods, results, discussion and conclusions, and even in references. Please, use “find and replace” function.

In reference #18 it is “invaded” indeed, because it refers to “invaded habitats”. But in references #20 and #21, it is not “invaded” but “invasive”. I checked the original titles and papers! You should have done it to!

Please, see here for example:

Glob Chang Biol. 2012 May; 18(5): 1725–1737.

doi:10.1111/j.1365-2486.2011.02636.x

“A global assessment of invasive plant impacts on resident species, communities and ecosystems: the interaction of impact measures, invading species' traits and environment”

By Petr Pyšek,*†‡ VojtÄ›ch Jarošík,*†‡ Philip E Hulme, Jan Pergl, Martin Hejda,* Urs Schaffner, and Montserrat Vilà||

Also, as I explained before there is no need to repeat words in keywords, words that make part of the tittle. This is redundant. Try to use different words in keywords that are not in the tittle.

In first line of “introduction” there is no need to write “(R. solanacearum)” after ”Ralstonia solanacearum”.

Regarding the objectives, as I stated, it should be in my opinion, (line 72) “evaluate its efficacy in managing a particular soil-borne disease”, not in general “soil-borne diseases”, as you tested only one particular and solely one disease, with only one strain of the bacteria R. solanacearum. The authors say that they followed my suggestion but they didn’t Please correct!

Moreover, in subsection 2.1 (“Planting materials and experimental soil”) the authors claim again that my suggestion was followed, but the error still continues. It is not “Planting materials…” but “Plant materials…

Lines 121 and 122: please rephrase.I think you mean “… they were used to be inoculated with the Tim 17 strain of R. solanacearum using the root injury method”.

Other minor things: in the tittle of subsections of the discussion (4.1 and 4.2), first word should begin with a capital letter.”

Line 321: correct “bacteral”: it is “bacterial”.

Line 330: It shoud be “Furthermore” and not ”Furthmore” as is wrongly written. Please correct.

Nevertheless, globally, my major concerns were addressed and issues were solved. I think that the paper still needs minor revision, as showed above, and afterwards it can be accepted. I would recommend revision of the English language and grammar, I’m afraid! (Because all those errors that were maintained).

Comments on the Quality of English Language

I would recommend revision of the English language and grammar, I’m afraid! Mainly, due to all those errors that were maintained. But if this is checked over in production, i am ok with it.

Author Response

Dear Prof.

Thank you very much for giving us an opportunity to revise our manuscript. We appreciate

the editor and reviewers very much for their constructive comments and suggestions on our

manuscript entitled “The win-win effects of an invaded plant biochar on a soil-crop system: Controlling a bacterial soilborne disease and stabilizing soil microbial community network”.

We have studied reviewers’ comments carefully. According to the reviewers’ detailed

suggestions, we have made a careful revision on the original manuscript. All revised portions are

marked in red in the revised manuscript which we would like to submit for your kind

consideration.

First of all, I must say that in spite of the authors state that my suggestions were taken into account, some errors still occur. For example, the authors keep using the expression “invaded plant” and they should use “invasive plant” (or “invading plant” or “plant invader” or “invasive alien plant” or “alien invader”). We are talking about an invasive plant (or plant invader), not a plant that is invaded! This error occurs all through the whole manuscript (36 times), namely in the tittle, keywords, introduction, materials and methods, results, discussion and conclusions, and even in references. Please, use “find and replace” function.

Response: Thank you very much for your detailed review and valuable comments. You pointed out an issue regarding the accuracy in the usage of "invaded" versus "invasive" in our references. We have thoroughly reviewed our manuscript and acknowledge the oversight. We agree that a more careful examination of the key terms and their correct application should have been conducted prior to submission. For this oversight, we extend our sincere apologies and appreciate your correction.

In reference #18 it is “invaded” indeed, because it refers to “invaded habitats”. But in references #20 and #21, it is not “invaded” but “invasive”. I checked the original titles and papers! You should have done it to!

Please, see here for example:

Glob Chang Biol. 2012 May; 18(5): 1725–1737.

doi:10.1111/j.1365-2486.2011.02636.x

“A global assessment of invasive plant impacts on resident species, communities and ecosystems: the interaction of impact measures, invading species' traits and environment”

By Petr Pyšek,*†‡ VojtÄ›ch Jarošík,*†‡ Philip E Hulme,‡ Jan Pergl,*§ Martin Hejda,* Urs Schaffner,¶ and Montserrat Vilà||

Response: Thank you for your suggestion. It has been replaced as suggested.

Also, as I explained before there is no need to repeat words in keywords, words that make part of the tittle. This is redundant. Try to use different words in keywords that are not in the tittle.

Response:  Thank you for your suggestion. We have reconsidered the keywords and replaced "invsive plant" with "Solidago canadensis L."

In first line of “introduction” there is no need to write “(R. solanacearum)” after ”Ralstonia solanacearum”.

Response: Thank you for your suggestion. It has been replaced as suggested.

Regarding the objectives, as I stated, it should be in my opinion, (line 72) “evaluate its efficacy in managing a particular soil-borne disease”, not in general “soil-borne diseases”, as you tested only one particular and solely one disease, with only one strain of the bacteria R. solanacearum. The authors say that they followed my suggestion but they didn’t Please correct!

Response: Thank you for your feedback. We acknowledge our mistake in not specifically stating our objective as "evaluate its efficacy in managing a particular soil-borne disease caused by R. solanacearum." We will correct this in line 72 to accurately reflect our study's narrow focus. We appreciate your guidance and will ensure our manuscript aligns with our actual research scope.

Moreover, in subsection 2.1 (“Planting materials and experimental soil”) the authors claim again that my suggestion was followed, but the error still continues. It is not “Planting materials…” but “Plant materials…

Response: Thank you for pointing out the inconsistency in the subsection title. We apologize for the oversight and any confusion it may have caused. We will correct the title from "Planting materials and experimental soil" to "Plant materials and experimental soil" to accurately reflect the content discussed and to adhere to your suggestion. We appreciate your attention to detail and are committed to ensuring the accuracy of our manuscript.

Lines 121 and 122: please rephrase.I think you mean “… they were used to be inoculated with the Tim 17 strain of R. solanacearum using the root injury method”.

Response: Thank you for your suggestion. It has been replaced as suggested.

Other minor things: in the tittle of subsections of the discussion (4.1 and 4.2), first word should begin with a capital letter.”

Response: Thank you for your suggestion. It has been replaced as suggested.

Line 321: correct “bacteral”: it is “bacterial”.

Response: Thank you for your suggestion. It has been replaced as suggested.

Line 330: It shoud be “Furthermore” and not ”Furthmore” as is wrongly written. Please correct.

Response: Thank you for your suggestion. It has been replaced as suggested.
